# Role of Ultrasound in the Assessment and Differential Diagnosis of Pelvic Pain in Pregnancy

**DOI:** 10.3390/diagnostics12030640

**Published:** 2022-03-05

**Authors:** Martina Caruso, Giuseppina Dell’Aversano Orabona, Marco Di Serafino, Francesca Iacobellis, Francesco Verde, Dario Grimaldi, Vittorio Sabatino, Chiara Rinaldo, Maria Laura Schillirò, Luigia Romano

**Affiliations:** Department of General and Emergency Radiology, “Antonio Cardarelli” Hospital, 80131 Naples, Italy; marco.diserafino@aocardarelli.it (M.D.S.); iacobellisf@gmail.com (F.I.); francescoverde87@gmail.com (F.V.); dariogrimaldi@me.com (D.G.); vittorio.sabatino@gmail.com (V.S.); chiara_rinaldo@libero.it (C.R.); marialaura.schilliro@gmail.com (M.L.S.); luigia.romano1@gmail.com (L.R.)

**Keywords:** gynaecology, obstetrics, gastrointestinal tract, urinary system, vascular system, magnetic resonance, emergency

## Abstract

Pelvic pain (PP) is common in pregnant women and can be caused by several diseases, including obstetrics, gynaecological, gastrointestinal, genitourinary, and vascular disorders. Timely and accurate diagnosis as well as prompt treatment are crucial for the well-being of the mother and foetus. However, these are very challenging. It should be considered that the physiological changes occurring during pregnancy may confuse the diagnosis. In this setting, ultrasound (US) represents the first-line imaging technique since it is readily and widely available and does not use ionizing radiations. In some cases, US may be conclusive for the diagnosis (e.g., if it detects no foetal cardiac activity in suspected spontaneous abortion; if it shows an extrauterine gestational sac in suspected ectopic pregnancy; or if it reveals a dilated, aperistaltic, and blind-ending tubular structure arising from the cecum in suspicious of acute appendicitis). Magnetic resonance imaging (MRI), overcoming some limits of US, represents the second-line imaging technique when an US is negative or inconclusive, to detect the cause of bowel obstruction, or to characterize adnexal masses.

## 1. Introduction

### 1.1. Background of Pelvic Pain in Pregnancy

During pregnancy, pelvic pain (PP) is a relatively common condition sustained by various causes and characterized by different timing and intensity of clinic presentation. Lower back pain and PP are common symptoms during pregnancy defined as “recurrent or continuous pain for >one week from the lumbar spine or pelvis” with a wide range of reported incidence (24–90%) [1,2]. Therefore, a “crampy” PP is expected in early pregnancy and related to rapid growth in uterus size, hormonal changes, and increased blood flow. In the first trimester, pregnant patients often complain of painful symptoms associated with normal findings on diagnostic examinations [3]. Acute PP in pregnancy means a different condition defined by sudden onset of abdominal pain with a particular intensity related to a wide variety of diseases, including disorders of the obstetric, gynaecologic, gastrointestinal, genitourinary, and vascular systems. This potentially dangerous acute condition does not occur in a specific preferential week of pregnancy and always represents diagnostic and therapeutic challenges [4]. The most important factor influencing this context is the presence of vague or altered clinical signs that are obscured by concurrent maternal physiologic and anatomic changes [4,5]. Moreover, during pregnancy, white blood cells count is typically elevated, achieving the range of 20–30,000 cells/μL at the end of the third trimester [6]. The hydronephrosis related to ureteral compression or the displacement of organs such as the appendix by the gravid uterus could confound the clinical presentation and make diagnosis difficult. Imaging has gained the fundamental role of making a subtle clinical presentation clear, allowing for a prompt diagnosis and treatment that are essential for the mother and the foetus’s well-being [4,6]. Based on their incidence, it is possible to identify some common and typical causes of PP in the first trimester of pregnancy and others peculiar to the second-third trimester of pregnancy. The first group includes obstetric causes such as early pregnancy failure and ectopic pregnancy and gynaecologic causes such as complications related to adnexal masses, ovarian torsion, and gynaecologic causes and leiomyomas [7]. In the second group, complications of placental abruption and uterine rupture are described. Other causes of PP in pregnancy are related to gastrointestinal, genitourinary and vascular systems, and may occur in each trimester of pregnancy [8].

### 1.2. Diagnostic Imaging in Pregnancy

The protection of both mother and foetus represents the first criteria to choose the most appropriate diagnostic tools in the assessment of emergency diseases during pregnancy [9]. To date, ultrasound (US) and magnetic resonance imaging (MRI) are the preferred imaging techniques as imposed by the well-known risks to the foetus derived from ionizing radiation exposure [4,10]. US is rapid, painless, extensive results, widely available and considered safe. This modality does not require intravenous contrast media administration and represents the first-line exam in pregnant patients (Table 1) [11,12,13,14]. However, it is necessary to be aware of US limitations such as the operator-dependency, the small field of view, and the altered body habitus of a pregnant woman [4].

MRI is the diagnostic tool with the most accurate and powerful diagnostic performance when compared to any other imaging technique, especially when considering the absence of ionizing radiation and administration of contrast media [15,16]. No reports document the detrimental effects of MRI on the foetus development. In recent years, there has been a growing interest in the use of MRI in abdominal emergencies during pregnancy, particularly in the diagnosis of appendicitis, the most common cause of acute abdomen in pregnancy [17,18]. The American College of Radiology does not recommend the use of gadolinium during pregnancy due to its passage through the placenta to the foetal circulation with excretion by the foetal kidneys into the amniotic fluid and indefinite persistence in this compartment [19]. No large studies are available on the use of gadolinium-based contrast agents during pregnancy to prove their adverse effects. Hence, the entity of potential risks for the foetus remains unknown [6]. Computed tomography (CT) may also accurately detect many causes of abdominal pain during pregnancy, but the use of ionizing radiation on a pregnant or potentially pregnant patient always requires a careful and prudential risk-benefit analysis [6,20]. Therefore, it is necessary to use an automatic exposure control system to reduce radiation exposure and dedicated protocols to minimise the dose as much as possible without reducing image quality [21,22]. The correct use of diagnostic tools is essential. US represents the first line and provides information that allows radiologists to avoid delayed diagnosis in an emergency setting, giving both the mother and the foetus a chance.

## 2. Obstetric Causes

### 2.1. Spontaneous Abortion

First-trimester spontaneous abortion occurs in 10% to 12% of clinically recognized pregnancies [23,24]. Patients may be asymptomatic or present vaginal bleeding and PP, which could be constant, intermittent, or crampy over the uterus or lower back [25]. A gestational sac may first be visualised on the TVS at 4.5–5.0 weeks of gestational age as a 2–3-mm rounded intrauterine fluid collection. The mean sac diameter (MSD) growth rate is 1.13 mm per day but is often variable [9]. In a pregnant woman with PP and vaginal bleeding during the first trimester, US is the initial diagnostic exam of choice since it may confirm an early pregnancy failure with high specificity if the foetal cardiac activity is not detectable when the embryonic pole is 5 mm in length or the pregnancy is known to be 6.5 weeks (Figure 1) [26]. Therefore, the US could give additional suspicious information for bad pregnancy outcomes such as irregular sac shape, the low position of the gestational sac, bradycardia, large sac without a yolk sac or embryo. In the literature, the reported normal range values correspond to a MSD up to 13 mm for visualisation of the yolk sac and 18 mm for visualisation of the embryo [27]. Furthermore, the sonographic ”empty amnion sign” is represented by the visualisation of an amniotic sac without concomitant visualisation of an embryo and is considered an important finding of failed pregnancy regardless of the MSD value [28]. The correlation between maternal serum level of β-HCG and sonographic findings can also indicate an early pregnancy failure. Indeed, a gestational sac should be visible when the β-HCG level is higher than 2000 mIU/mL, while the embryo when the β-HCG level is higher than 10,800 mIU/mL [29].

### 2.2. Ectopic Pregnancy

Ectopic pregnancy (EP) represents the first cause of maternal death in the first trimester. The incidence has steadily increased in the last fifty decades proportionally to an increased prevalence of risk factors, such as assisted reproductive techniques and sexually transmitted diseases, particularly chlamydial infection [30]. In EP, pregnancy implantation occurs outside the uterine cavity and is differentiated into tubal and no-tubal EP. In particular, tubal EP is more frequent with an incidence of 95%. On the other hand, non-tubal implantation mainly affects mortality and occurs in 5% of EPs with localization in the uterine interstitium (cornual or angular), cervix, ovary, and previous Caesarean section scar. Furthermore, a third type of EP which should be considered is the heterotopic pregnancy, defined by the co-existence of intra-uterine and extra-uterine pregnancies. Heterotopic pregnancy shows a very low incidence in the general population (1 in 7000) with an increased rate after in vitro fertilization, reaching a value of 1% in some studies [31]. Clinical presentation varies; some patients report no symptoms while others show massive intra-abdominal haemorrhage and collapse, presenting as a life-threatening emergency. Classically, patients present amenorrhoea, followed by vaginal bleeding and PP. The presence of shoulder tip pain is considered an indirect sign of intra-abdominal bleeding, the irritation of the diaphragm causes referred shoulder pain [32]. In 90% of EP the diagnosis may be reliably reached using TVS as a single stand-alone test. Although the regular fallopian tubes are not visualised by US, pathological distension of the tubes can be clearly visualized, and 95% of EPs are tubal. In an US, the corpus luteum is classically described as a “ring of fire” on colour Doppler; it may be seen on the ipsilateral side of tubal EP in 70–85% of cases and represents an important marker [33]. An inhomogeneous mass (“blob sign”) is detected adjacent to the ovary in 60% of EPs, while a hyperechoic ring (“bagel sign”) is seen in 20% of cases. In the remaining 13% of cases, a gestational sac with a foetal pole appears, and a cardiac activity could be present or not (Figure 2) [34]. US diagnosis of non-tubal EPs could be suspected on the presence of trophoblastic tissue outside the endometrial cavity and surrounded by a thin myometrial plane in case of interstitial EP, whereas a barrel-shaped cervix with gestational sac below the level of the uterine arteries is indicative of cervical EP [35,36]. Furthermore, the differential diagnosis between ovarian EP and ovarian germ cell tumours may be challenging. However, the combination of US findings and high levels of serum hCG are very suspicious. Almost one-third of women have haemodynamic instability due to rupture [37]. Approximately 19% of women have a defect in the anterior myometrium at the level of the previous caesarean section, and the US detection of an empty endometrial cavity with a gestational sac in the anterior lower myometrium is characteristic of the Caesarean scar. In the case of caesarean scar implantation, the rarest type of EPs, the risk of uterine rupture, haemorrhage and hysterectomy is high [38,39]. The presence of liquid in the pouch of Douglas (POD) should always be evaluated and, if blood is present, it shows a ground-glass appearance, and a complementary transabdominal scan should be performed to inspect Morison’s pouch: the presence of blood at this level indicates significant intra-abdominal bleeding (equates to a minimum of 670 mL) [33].

### 2.3. Placental Abruption

The exact aetiology of placental abruption is unknown. Placental abruption is a relatively rare condition and accounts for 10–25% of prenatal deaths. For this reason, a prompt diagnosis and correct management are required. Most placental abruptions occur before 37 weeks of gestation [40]. This condition is defined by the separation of placenta from the myometrium in the uterus, the rupture of maternal vessels tear, and the consequent blood accumulation pushing the uterine wall away from the placenta. Symptoms range from asymptomatic condition to maternal shock, but vaginal bleeding and abdominal or PP often occur [41]. US is the first modality to evaluate placental abruption, and sonographic findings are usually related to the presence of hematomas. The echogenicity of hematomas depends on their age. Acute hematomas tend to be hyperechoic or isoechoic compared to the adjacent placenta, whereas sub-acute or chronic hematomas are commonly isoechoic to the placenta, and they might be misdiagnosed for placental focal thickening (Figure 3). A ‘normal’ ultrasound does not exclude a placental abruption [42]. In the majority of cases, hematomas are subchorionic (between the chorionic membrane and uterine wall), less frequently retroplacental (behind the placenta) and preplacental (in front of the placenta) [4].

### 2.4. Uterine Rupture

In case of uterine rupture, the abdominal pain is severe. Separation of all layers of the uterine wall with communication between the uterine cavity and the peritoneum occurs [43]. Previous uterine surgery, including caesarean deliveries and myomectomy, as well as congenital uterine malformations, are considered the most important risk factors [44]. Although the choice of the imaging technique depends on patient haemodynamic stability, US is often the first choice, capable of detecting indirect signs of uterine wall injury such as gas within the uterine defect and hemoperitoneum. The direct US findings are represented by a bulky empty uterus with anechoic line, which corresponds to uterine tear, or the dislocation of foetus and placenta in the abdominal cavity associated with increased free fluid (Figure 4) [44,45].

## 3. Gynaecologic Causes

### 3.1. Adnexal Masses or Ovarian Cyst

Adnexal masses occur in approximately 1–5.3% of all pregnancies, and they are generally asymptomatic [46]. Their incidence has increased in recent decades due to the large use of the US during pregnancy. In most cases, the ovarian mass is a simple cyst destined to disappear spontaneously during pregnancy; a minor percentage persists in the second and third trimesters and requires imaging follow-up or surgical removal [47,48]. The most common adnexal masses during pregnancy are represented by corpus luteum (often haemorrhagic) or other functional ovarian cysts. Other types of cysts can also rupture, including cystic neoplasms. The most commonly ruptured cystic neoplasms are mature cystic teratomas [49]. Although the adnexal masses in pregnancy tend to be benign, it is essential to remember that ovarian cancer still represents the second most common gynaecological neoplasm during pregnancy after cervical cancer [50]. US examination provides to identify the main characteristics of the mass such as dimension and signs of complexity with a higher likelihood of malignancy [51]. The dimensional increase of the uterus could determine compression on the adnexal mass or adjacent organs on the adnexal mass. The compression could be complicated by torsion, haemorrhage, or rupture [52]. Generally, MRI is used as the first-line imaging technique to characterize adnexal masses in order to define the best patient management [53].

### 3.2. Adnexal Torsion

The incidence of ovarian torsion in the general population is not well known, but it results to be higher during pregnancy, probably due to the displacement of the adnexa out of the pelvis, especially during the first and early second trimester [47]. Adnexal torsion is defined as the rotation of the ovarian peduncle around its axis, which may determine ischemia and ovarian necrosis. It can involve either the ovary or the fallopian tubes if torsion of the infundibulo-pelvic and tubo-ovarian ligaments occurs. The torsion of the vascular peduncle causes impaired lymphatic and venous outflow, resulting in enlargement and widespread ovarian oedema. The arterial flow can initially be preserved since the arteries have thicker muscle walls and are less prone to collapse. However, if not treated, arterial thrombosis, ischaemia, or necrosis of the ovaries occur [54]. Timely diagnosis and surgery are crucial for preserving ovaries. Typically, patients present with a sudden onset of acute and intense abdominal or PP and additional symptoms may include nausea, vomiting, flank pain, and fever with mild leukocytosis. On US examination a one-sided enlarged ovary, usually larger than 4 cm, with or without associated mass is visualised; further US findings are represented by the presence of multiple small ovarian follicles positioned peripherally on the enlarged ovary with hypoechoic central edema change or central ovarian mass, if present (i.e., the ‘pearl necklace’ appearance) [55]. In addition to the grayscale findings, colour and spectral Doppler evaluations are important for assessing adnexal torsion. As with grayscale imaging, Doppler findings vary according to the degree of torsion, the time elapsed since the onset of the disease, and the degree of vascular compromise. The absence of detectable blood flow in the affected ovary allows a confident diagnosis of torsion with a positive predictive value of 94% [56]. However, several studies have shown that the detection of flow within an ovary, using colour Doppler and spectral Doppler, cannot rule out the diagnosis of torsion. The double arterial flow that supplies the ovary can help maintain arterial flow, even if the initial loss of venous flow occurs. This means that whether the torsion is early, intermittent, or partial, both venous and arterial flow may be preserved [54,55,56]. The analysis of spectral Doppler waveforms can increase the sensitivity of torsion diagnosis. An arterial waveform with reversal of diastolic flow, indicating high resistance, can suggest the diagnosis of torsion [55]. However, in conclusion, the grayscale findings associated with the clinical signs are considered more reliable than Doppler findings in diagnosing adnexal torsion. A twisted vascular pedicle or vortex sign is a valuable finding in both grayscale and Doppler for diagnosing adnexal torsion [56].

### 3.3. Uterine Leiomyoma

Leiomyomas are benign neoplasms composed of smooth muscle and varying amounts of fibrous tissue. They can originate anywhere in the uterus and can occur singularly or in multiples, with sizes ranging from a few millimetres to several centimetres. Based on their location, they are intramural (the majority), submucosal (including intracavitary), or subserosal [57]. Approximately half of all leiomyomas grow during pregnancy, especially in the first trimester due to the increased estrogen levels [58]. The rapid growth can suffer from an insufficient vascular supply that determines degeneration and necrosis and results in severe abdominal pain and uterine contractions. “Red degeneration” is the most common type of degeneration during pregnancy. This kind of degeneration occurs when the rapid growth of leiomyomas exceeds blood supply with consequent bleeding [59]. On US, small leiomyomas are usually homogeneous, while those with a diameter greater than 3 cm tend to be heterogeneous (Figure 5). In detail, when leiomyomas increase in size, they tend to outgrow their vascular supply, and the hyaline, myxoid, cystic, or haemorrhagic degeneration may occur [60]. The leiomyoma may present a more atypical appearance on US in these cases. Furthermore, the degeneration can lead to edema, which in turn can lead to the formation of cystic spaces, echogenic haemorrhagic areas, and dystrophic calcification [57,61].

### 3.4. Endometriosis

Endometriosis is defined as the presence of endometrial glands and stroma in ectopic areas outside the uterus, and symptoms are often associated with recurrent bleeding, dysmenorrhoea, dyspareunia, and chronic PP [62]. The most common endometriotic implantation sites include the ovary’s surface, the suspensory ligaments of the uterus, the uterus, the peritoneal surfaces of the Douglas cavity, and the fallopian tubes. Endometriotic implants can also occur in the anterior abdominal wall (extra-pelvic endometriosis), typically near a surgical scar, at the entry point of a needle, laparoscopic trocar or Caesarean section [63,64]. Although the pain is usually chronic, some complications can also result in episodes of acute pain during pregnancy [65]. Pregnancy was usually believed to positively affect endometriosis and its painful symptoms due to metabolic, hormonal, immune, and angiogenic changes. Recently, an emerging role of endometriosis has been defined in affecting the development and outcome of pregnancy [66]. Spontaneous hemoperitoneum, bowel and ovarian complications are unpredictable and, although rare, they represent life-threatening conditions that need surgical intervention in most cases. Some evidence demonstrated a correlation between endometriosis and spontaneous miscarriage, preterm birth, small for gestational age babies and placenta previa [66]. US findings of endometriomas may coincide considerably with other adnexal masses, including haemorrhagic cysts, tubo-ovarian abscess (TOA), dermoid cysts, and ovarian cyst-related neoplasms [52]. US represent the first-line imaging exam used to identify a solid, heterogeneous, and hypoechoic mass with diffuse internal echoes, even if the endometriosis US findings are variable. Cysts can change during pregnancy; the margins may become not well-defined and infiltrate adjacent soft tissues. Most implants show vascular flow on Doppler. Differential diagnosis includes masses on abdominal walls, such as desmoid tumour, metastasis, lymphoma, melanoma, hematoma, suture granuloma, or scar hernia [67,68,69]. Haemorrhagic and dermoid cysts are the two entities with which the differential diagnosis is more complicated. US follow-up is performed at short intervals since haemorrhagic cysts tend to resolve spontaneously, while endometriomas will tend to persist. If US findings are not clear and ovarian cancer is suspected, MRI is mandatory [70].

### 3.5. Pelvic Inflammatory Disease

Pelvic inflammatory disease (PID) refers to a spectrum of conditions that occur when microorganisms ascend from the lower genital tract to the uterus, fallopian tubes, and ovaries and is a common cause of referral to the emergency department and hospitalisation for acute gynaecological disorders [71]. PID during pregnancy is not commonly reported. Unlike the pelvic abscess, potentially discovered at any gestation age, acute salpingitis occurs more commonly in the first trimester. Both processes are associated with loss of embryo or foetus through spontaneous abortion or stillbirth [72,73]. Regardless of pregnant status, the infection continuum begins with cervicitis and progresses to endometritis, salpingitis, pyosalpinx, tubo-ovarian complex, and TOA. Chlamydia trachomatis or Neisseria gonorrhoeae are responsible for one-third to one-half of cases [74]. Although salpingitis may not always be recognisable on US, infected fallopian tubes can show thickened and hyperaemic walls. Occlusion of the ovarian fimbria of the fallopian tubes due to inflammation will result in a dilated tuba containing pus or in a pyosalpinx, which appears as a dilated tubular structure with echogenic intraluminal fluid and debris, sometimes with layered echoes indicating the presence of liquid and sediments. The presence of thick, hyper-vascularised walls suggests acute disease [71]. The final phase is the formation of the TOA, in which the ovary is no longer recognisable, and an inflammatory mass covers both the ovary and the fallopian tube. Rupture of a TOA can result in septic shock [52,71].

## 4. Urinary Tract Causes

The urinary tract causes of abdominal pain during pregnancy are represented by obstructive hydronephrosis as well as infectious diseases. First of all, the physiological changes to the urinary tract during pregnancy must be discussed. In the first and second trimesters, the glomerular filtration rate increases by 40–65% due to the rise in cardiac output, total vascular volume, and renal blood flow. Consequently the kidney volume increases [75]. Furthermore, in the second trimester, a “physiological hydronephrosis” occurs in more than half of pregnancy and is due to the combination of high levels of progesterone and gonadotrophin, which induce smooth muscle cell relaxation and the extrinsic compression by the growing uterus and enlarged ovarian veins against iliopsoas muscle (Figure 6) [76]. This finding is more common on the right side due to the dextrorotation of the gravid uterus. On the left side, the sigmoid colon “protects” the ureter. During the first eight weeks after delivery, the physiological hydronephrosis usually disappears.

In flank or PP during pregnancy, this physiological change should be considered and differentiated from obstructive hydronephrosis. In particular, obstructive hydronephrosis is usually caused by urinary tract calculi and is more common in the second or third trimesters. The right and left sides are equally involved, and the incidence varies from 1 in 90 to 1 in 3800 pregnancies [77]. The prompt and right diagnosis is necessary to prevent potential complications, such as pyelonephritis and premature labour, for that reason hospitalization is often required, although in most cases, conservative management is the preferred approach. US represents the first-line imaging technique in assessing flank and PP to make the differential diagnosis between acute renal obstruction, generally by urolithiasis, and other non-urinary causes, such as appendicitis, diverticulitis, placental abruption, or primary premature labour. The incidence of urolithiasis during pregnancy is no different from that of non-pregnant women and is low (0.03–0.6%) [78]. Furthermore, mild or severe pain may be caused by compression of the ureter by the gravid uterus. Hence, when a dilatation of the collecting system is observed in a pregnant woman, as mentioned above, the first thing to do is to differentiate between obstructive and physiological hydronephrosis. In detail, in case of physiological hydronephrosis, the upper cavities are dilated as well as the lumbar ureter, while the pelvic portion is always normal. Unfortunately, only calico-pelvic cavities are usually examinable at US, while ureter is not due to the gravid uterus and overlying bowel gas, however its visualization increases with the grade of hydronephrosis. The most important point to be evaluated is the level where the lower lumbar ureter crosses the common iliac artery since, in case of physiologic dilatation, the ureter is tapered at this site and not dilated below. The integrated assessment with colour Doppler aid in differentiating the ureter from iliac vessels and enlarged ovarian veins. In case of urolithiasis, the stone may be identified by anterolateral approach if it obstructs the mid or lower lumbar ureter, while by anterior transabdominal or endovaginal approach if it is located in the lower pelvic or terminal portions. Furthermore, colour Doppler aids in the identification of small pelvic stones by twinkling artifacts. Since ureters are not always visible on US, secondary findings are necessary to evaluate. In literature, authors have proposed cavity measurements to differentiate obstructive and physiologic hydronephrosis, but they are not used in clinical practice, and there is no consensus [79]. Therefore, qualitative changes are considered the most important findings in the differential diagnosis. In particular, the absent or slight dilatation on the symptomatic side exclude pathological obstruction, while a predominant left dilatation with left pain is suggestive of pathological obstruction. The measurement of intrarenal resistivity index (RI) may aid in the diagnosis: it increases (>0.7) within six hours in case of acute and complete ureteral obstruction and has a diagnostic value only when positive. A difference of 0.04 or greater between normal and abnormal kidney RI is an accurate indicator with high sensitivity, specificity and accuracy (95%, 110%, 99%, respectively) (Figure 7) [80,81].

Another finding which may aid in the diagnosis of obstructive dilatation is the assessment of the ureteral jet for at least 1 min. If urinary jet is present on the symptomatic side and symmetrical with the contralateral side, significant obstruction is unlikely, whereas if it is unilaterally altered, even in the contralateral oblique decubitus position to decrease uterus mass effect on the bladder and ureter, complete obstruction is highly probable with a reported sensitivity of 100% and specificity of 91% (Figure 8 and Figure 9) [80]. US may direct show calculi with a sensitivity ranging between 34% and 95% or reveal secondary findings of acute obstruction, such as perinephric fluid or absence of ureteral jet [80]. In most cases, stones are smaller than 5 mm and will pass spontaneously with analgesia, bed rest, and hydration.

When symptoms continue despite conservative management or when US is negative, MR urography is the second-level imaging technique [16]. Stones appear as signal voids overlying the high signal of urine within a dilated ureter. MR findings suspicious of obstructive hydronephrosis are represented by unusual sites of obstruction, such as the ureteropelvic junction, an abrupt ending of the ureter and perinephric or periureteral edema. On the other hand, physiologic hydronephrosis is characterized by gradual, smooth tapering of the mid to distal ureter. Finally, CT remains the last chance in unresolved cases to detect urinary tract calculi. The low-dose protocol exposed foetus to an average estimated dose of 7 mGy, below the 50-mGy limit above which there is a statistically higher risk of teratogenesis [82]. One of the most important complications of obstructive hydronephrosis is represented by pyelonephritis. The clinical presentation consists of high fever and flank pain, blood tests reveal leucocytosis and increased C-reactive protein. Due to lack of ionizing radiation, US is the first imaging technique in the suspicion of pyelonephritis, but, unfortunately, it has low sensitivity, showing abnormalities in only 25% of cases [83]. Possible findings are the presence of debris in the collecting system, areas of reduced cortical vascularity on power Doppler, and abnormal echogenicity of the renal parenchyma due to edema consisting of focal/segmental hypoechoic regions (Figure 10).

MRI is more sensitive than US in the assessment of pyelonephritis without administration of contrast medium. Areas of focal pyelonephritis are characterized by lower signal intensity on T2-weighted sequences and restricted proton diffusion on DWI [84,85]. Furthermore, US may be useful in assessing local complications, such as abscesses, which appear as a well-defined hypoechoic area within the cortex or in the corticomedullary parenchyma, and perinephric collections, which may show hypoechoic or heterogeneous echotexture. Pyelonephritis in pregnancy is more common than in nonpregnant state and require special attention since they are associated with preterm birth and low birth weight. In the case of pyelonephritis during peri-partum, CT should be performed after delivery. On pre-contrast-phase kidneys may appear normal or oedematous and calculi or gas may be detected. On post-contrast phases, one or more focal wedge-like regions demonstrated reduced enhancement compared with the normal parenchyma (Figure 11) [86].

Furthermore, another urinary tract cause of preterm labour is represented by haemorrhagic cystitis. In the assessment of cystitis, imaging plays a minor role, since clinical symptoms and the results of midstream urine culture are usually enough to initiate treatment. However, US may show a diffuse wall thickening with prominent hypervascularity on colour Doppler and the bladder contents may be turbid due to debris or blood products in case of haemorrhagic cystitis.

## 5. Gastrointestinal Causes

The gastrointestinal causes of PP during pregnancy include appendicitis, inflammatory and obstructive processes of the bowel, which are not unique in pregnancy, but may be more challenging to diagnose. US and MRI are the preferred imaging techniques since they do not use ionizing radiation, unlike CT.

### 5.1. Appendicitis

Acute appendicitis is a typical surgical emergency during pregnancy and accounts for 1 in 1.500 deliveries [87]. It requires an early diagnosis since perforation increases during pregnancy with a resulting increased rate of foetal loss and maternal mortality [6]. The clinical diagnosis is more difficult due to the variable appendiceal position: the appendix is gradually displaced upward during pregnancy and, after the first trimester, women may complain of acute right upper quadrant abdominal pain [88]. Furthermore, the gravid abdomen limits the physical examination and symptoms, such as nausea, vomiting, and leucocytosis, are not specific. US is the imaging technique of choice, and the sonographic criteria for diagnosing appendicitis in pregnant patients are the same as in nonpregnant ones: a dilated (>6–7 mm) aperistaltic, non-compressible, and blind-ending tubular structure arising from the cecum [89]. Other findings may be associated, such as wall thickening (>2 mm), appendicolitis, surrounding hyperechoic inflamed fat, or hypoechoic fluid (Figure 12). It has to be highlighted that an elevated or retrocecal appendix may be difficult to assess on US, hence a negative US examination associated with high clinical suspicion for appendicitis requires the performance of additional imaging, such as MRI [16]. This technique shows high sensitivity (100%) and specificity of 94% [90]. If MRI is not available or contraindicated, CT is necessary to prevent delayed diagnosis and treatment. The risk of misdiagnosis outweighs the small potential risk of ionizing radiation [91].

### 5.2. Inflammatory Bowel Disease

The inflammatory bowel disease (IBD) disease activity is independent of pregnancy, but the activity is correlated with increased foetal loss rate and foetal growth retardation [92]. During pregnancy, US represents the first imaging technique to assess abdominal pain and, in case of IBD, it may show a thick-walled bowel segment (3–4 mm), which might be expression of active inflammation. The affected loop is often non-compressible and ipoperistaltic, has no mural stratification, and a hyperechoic, circumferential layer external to the bowel wall may be present and represents fibrofatty proliferation, sign of active inflammation. Furthermore, three or more mesenteric lymph nodes may be present as well as free intraperitoneal fluid [93,94]. In order to evaluate accurately the extent of the disease as well as any complications, such as bowel obstruction, fistulas or abscess formation, cross-sectional imaging is required, and MRI is preferred. In detail, this imaging technique shows thick-walled segments, bowel stenosis, fibrofatty proliferation, mesenteric adenopathy, and fistulas [95,96].

### 5.3. Bowel Obstruction

Bowel obstruction is the third gastrointestinal emergency during pregnancy, with an incidence of 1 in 2.500 deliveries and the most frequent cause is adhesions (60–70%), followed by volvulus (25%) [4]. This disease occurs more commonly in the third trimester, maybe due to increased mass effect of the gravid uterus on the small and large intestine. The clinical diagnosis is challenging since physical examination is limited by enlarged uterus and some symptoms, such as nausea, vomiting, and abdominal pain, are also present during pregnancy [92]. However, the onset of these symptoms after the first trimester has to raise the suspicious of intestinal disease. As usual in pregnancy, US is the first-line imaging technique and may show dilated, aperistaltic loops of bowel with fluid levels, but it does not reveal the point or cause of bowel obstruction [4]. For that reason, cross-sectional imaging is required and MRI is preferred to CT [16]. If it cannot be performed, CT is mandatory since the risk of delayed diagnosis and treatment of a bowel obstruction expose the foetus to a greater risk than radiation exposure.

## 6. Vascular Causes

Among vascular causes of PP, which have a higher incidence in pregnancies, venous thromboembolic disease and gonadal vein dilatation have to be mentioned.

### 6.1. Thrombosis of the Gonadal Veins

The risk of venous thrombosis increases during pregnancy due to venous stasis and hypercoagulability [6]. The first begins in the first trimester, peaks at around 36 weeks of gestation, and results from progesterone-induced venodilation, pelvic venous compression by the gravid uterus, and pulsatile compression of the left iliac vein by the right iliac artery. The latter is due to the progressive activation of the haemostatic system for the haemostatic challenge of delivery. The majority of venous thromboembolic events occur in the lower extremities, but pelvic, hepatic, mesenteric, and gonadal venous thrombosis are more frequent during pregnancy. In particular, gonadal vein thrombosis occurs in 80–90% of cases in the right vein due to higher pressure than the left one, and predominantly affects women in the puerperium, following less than 0.05% of natural births and 1–2% of caesarean delivery cases. It is often associated with gynaecological malignancy, pelvic surgery, and pelvic inflammatory disease (PID) [97]. Patients generally complain of acute lower quadrant pain, fever, and leucocytosis; a prompt diagnosis and therapy are required since it may result in pulmonary embolism, thrombus extension into the inferior vena cava and renal veins, or ovarian infarction [98]. Even if the role of US is often limited for this pathology since the ovarian veins are difficult to visualize in the presence of bowel meteorism and gravid uterus, it represents a first-level imaging technique and is useful for making differential diagnosis with appendicitis and ovarian torsion. When the ovarian vein is assessable on US, it appears as a tubular or serpentine avascular hypoechoic/anechoic structure, located superiorly to the ovary and anteriorly to the psoas muscle. The extension into the inferior vena cava or renal vein should be evaluated. US Doppler shows decreased or absent flow in the ovarian vein, depending on partial or total occlusion [98]. CT and MRI are more accurate in the diagnosis of gonadal vein thrombosis. In particular, MRI is preferred during pregnancy since it does not use ionizing radiation or contrast medium. Indeed, nonenhanced MR venography performs with the phase-contrast or time-to-flight technique can depict veins with flowing blood and occlusion sites [99]. On CT, the thrombosed ovarian vein appears as a tubular structure with an enhancing wall and low-attenuation lumen due to thrombus (Figure 13) [98].

### 6.2. Gonadal Vein Syndrome

Enlargement of the right gonadal vein in the late second and third trimesters of pregnancy is common in imaging studies in patients with right-sided abdominal pain. Extrinsic compression of the ureter and consequently the enlargement of the right gonadal vein has been proposed as the cause of pain. However, it should be emphasized that it must be an exclusion diagnosis when the right gonadal vein enlargement is the only abnormal imaging finding, and further studies to assess the relationship with PP are required [6,90].

## 7. Conclusions

Determining the cause of PP during pregnancy is challenging since physiological changes occur. Free ionizing-radiation imaging techniques should be preferred. In this setting, US plays a pivotal role due to its lack of ionizing radiation and widely availability. Hence, radiologists need to develop good expertise in this field. MRI overcomes some US limitations, such as the small field of view and the presence of interfering overlying structures, but it is not always available. In unresolved cases, CT remains a reliable technique. Low-dose protocols are mandatory; usually they expose foetus to an average estimated dose below the 50-mGy limit (above which there is a statistically higher risk of teratogenesis).

## Figures and Tables

**Figure 1 diagnostics-12-00640-f001:**
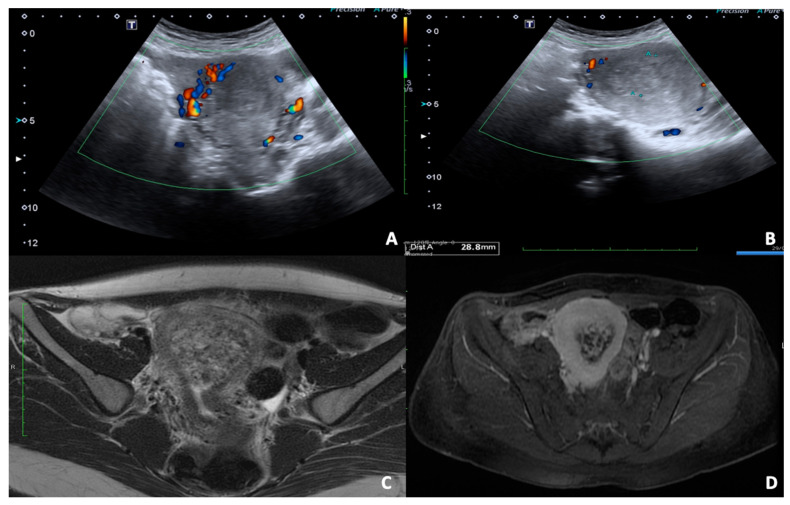
36 y.o. woman with pelvic pain and vaginal bleeding during the first trimester. (**A**,**B**) Convex probe axial and longitudinal scans of uterus show the presence of abnormal uterine shape for inhomogeneous content of cavity without concomitant visualisation of an embryo; the foetal cardiac activity was not detectable. (**C**,**D**) T2 FSE and TS fat-sat axial images of enlarged uterus confirm the altered content of cavity with no recognisable fetus.

**Figure 2 diagnostics-12-00640-f002:**
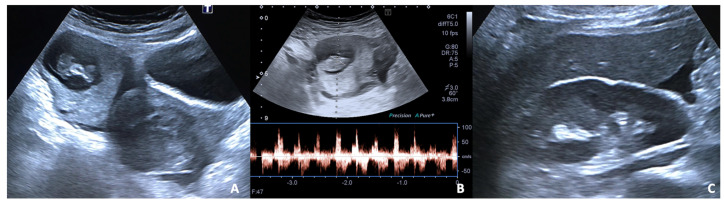
27 y.o woman with vaginal bleeding and pelvic pain. (**A**) US examination shows an inhomogeneous mass (“blob sign”) morphologically similar to a gestational sac in the right ovary; (**B**) cardiac activity is detected on power doppler; (**C**) free fluid is also present at the hepato-renal interface.

**Figure 3 diagnostics-12-00640-f003:**
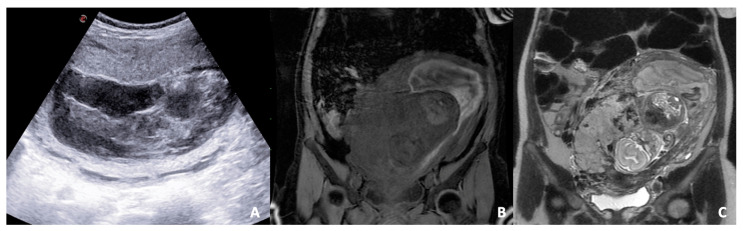
B-mode US scan (**A**) and T1 LAVA (**B**) and T2 FSE (**C**) coronal images of a pregnant woman. (**A**) US reveals a severe change of placental structure with inhomogeneous placental thickening. (**B**,**C**) MRI sequences show the separation of the placenta from the myometrium in the uterus due to blood accumulation and compression on the uterine wall away from the placenta; a subchorionic hematoma (localised between the chorionic membrane and uterine wall) is diagnosed.

**Figure 4 diagnostics-12-00640-f004:**
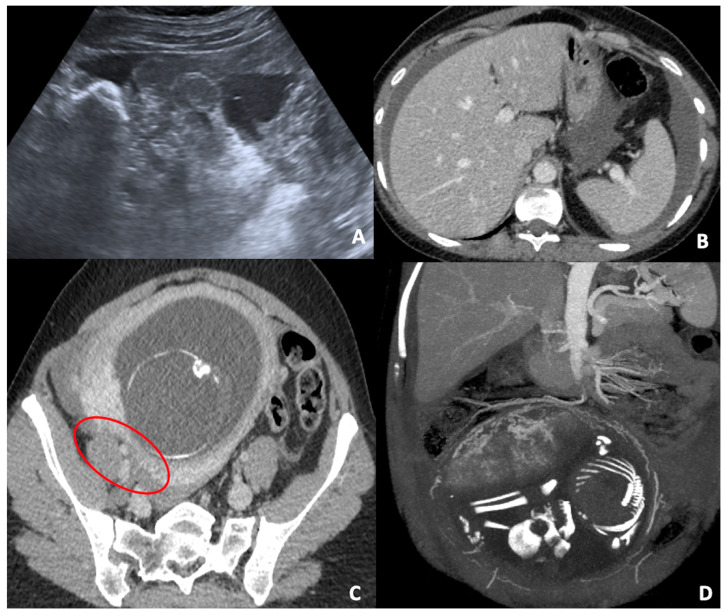
(**A**) Pelvic US scan of free inhomogeneous fluid in the pouch of Douglas compatible with hemoperitoneum; the uterus is enlarged and inhomogeneous with apparent wall defect. (**B**,**C**) CT axial images of the same patient show conspicuous hemoperitoneum; the gravid uterus is characterized by a severe thickness wall reduction (red circle), which corresponds to uterine tear. (**D**) CT coronal post-processed image (MIP) in venous phase clarifies the uterine vascularization and bleeding site.

**Figure 5 diagnostics-12-00640-f005:**
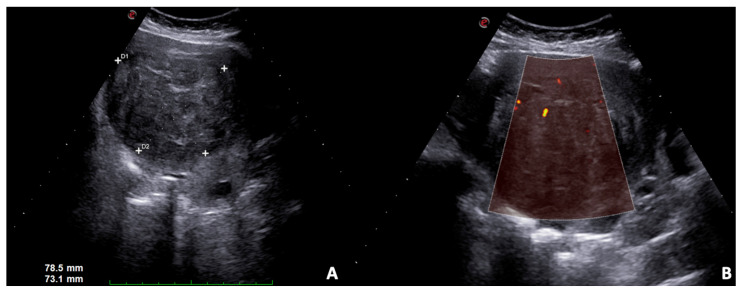
US B-mode scans. (**A**) Large solid leiomyoma of the uterus with inhomogeneous internal echotexture, (**B**) some vascular spots at power Doppler.

**Figure 6 diagnostics-12-00640-f006:**
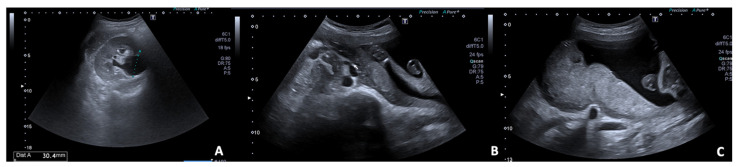
US B-mode scans in a pregnant woman during the third trimester. (**A**) Axial image of the right kidney shows a regular cortical-medullary pattern and physiological dilatation of renal pelvis; (**B**) also the lumbar tract of the right ureter is dilated. (**C**) Exploration of pelvis detects increase in size of the uterus with compression effect on the surrounding structures.

**Figure 7 diagnostics-12-00640-f007:**
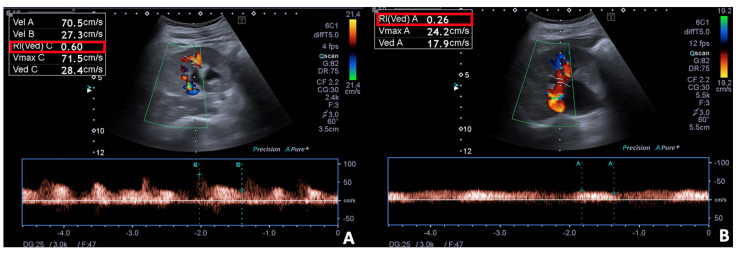
Spectral Doppler and Colour Doppler scans of the right kidney in a pregnant woman with urinary obstruction. Presence of moderate hydronephrosis without the US detection of a calculus; (**A**) the resistivity index is normal, (**B**) while the venous impedance index is increased.

**Figure 8 diagnostics-12-00640-f008:**
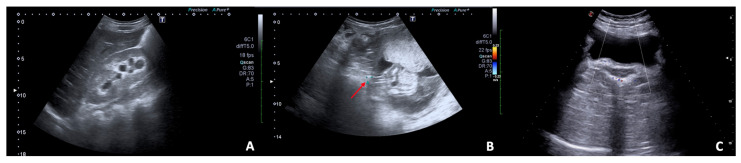
US B-mode (**A**,**B**) and Colour Doppler (**C**) scans in pregnant woman with right flank pain and nausea. (**A**) Collecting system of the right kidney is dilated with evidence (**B**) of a calculus (arrow) in the middle tract of the lumbar ureter resulting in complete urinary obstruction and mild hydronephrosis; (**C**) no right ureteral jet is detected on Colour-Doppler.

**Figure 9 diagnostics-12-00640-f009:**
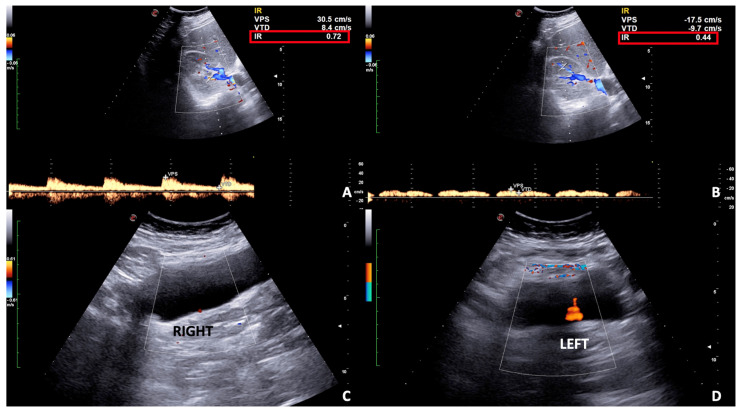
Spectral (**A**,**B**) and Color (**C**,**D**) Doppler scans of the right kidney and bladder in a pregnant woman with right flank and pelvic pain in incomplete urinary obstruction. (**A**) Mild hydronephrosis with no notable obstructive calculus is detected and the resistivity index is slightly increased on spectral Doppler (**A**) as well as the venous impedance index (**B**). (**C**,**D**) The ureteral jet is reduced on the right compared to the left side.

**Figure 10 diagnostics-12-00640-f010:**
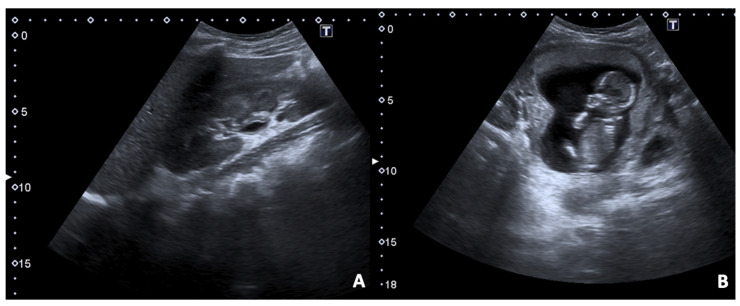
US B-mode scans in a pregnant woman arrived in the ER presenting fever and dysuria associated with right flank and pelvic pain. (**A**) A longitudinal image of hepato-renal scan shows thickened wall of right renal pelvis due to inflammation; (**B**) in the pelvis the foetus within the gravid uterus is demonstrated.

**Figure 11 diagnostics-12-00640-f011:**
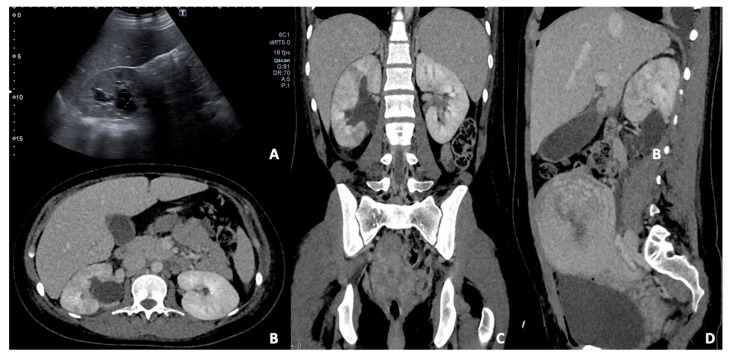
(**A**) US B-mode scan in a pregnant woman diagnosed with pyelonephritis and moderate dilatation of the right renal pelvis. Contrast-enhanced CT axial (**B**), coronal (**C**), and sagittal (**D**) images in the venous phase of the same patient after delivery. The right kidney appears slightly enlarged than the left one with persistent pelvis dilatation; some focal wedge-like regions of reduced enhancement are detected, confirming the presence of peri-partum pyelonephritis.

**Figure 12 diagnostics-12-00640-f012:**
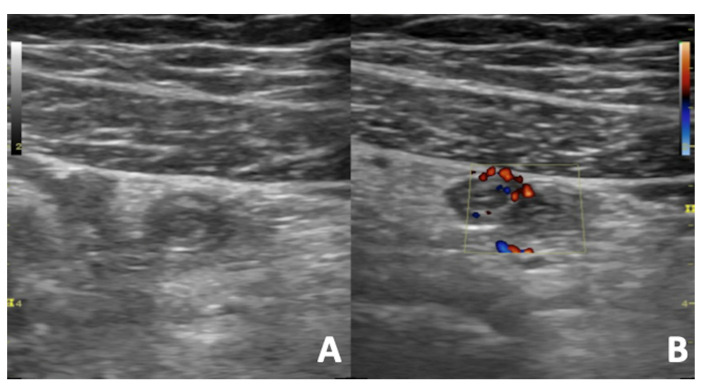
US B-mode (**A**) and Colour Doppler (**B**) scans of a pregnant woman with right iliac fossa pain. Acute appendicitis appears as aperistaltic, non-compressible and blind-ending tubular structure with thickened walls arising from the cecum. The surrounding fat is hyperechoic and inflamed. The color Doppler reveals an increased vascular flow of the appendix walls due to inflammation.

**Figure 13 diagnostics-12-00640-f013:**
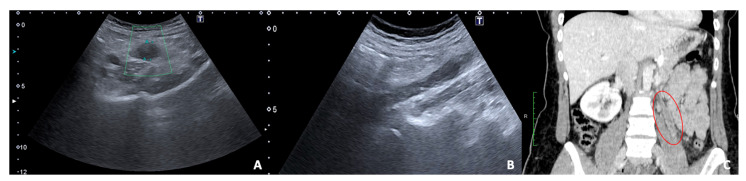
US and CT examinations of a pregnant woman who suffered from left pelvic pain before delivery. US B-mode scans of the left ovarian vein in axial (**A**) and longitudinal (**B**) views before delivery show the ovarian vein as a tubular structure with heterogeneous hypoechoic echotexture, located superiorly to the ovary and anteriorly to the psoas muscle. (**C**) Contrast-enhanced CT coronal image was performed after delivery and confirmed the left gonadic vein thrombosis (red circle).

**Table 1 diagnostics-12-00640-t001:** Sonographic methods, protocol, utility, and limits for examining the female pelvis.

Sonographic Methods	Probe	Protocol	Utility	Limits
**Transabdominal sonography (TAS)**	Low-frequency probe (convex probe 3–5 MHz)	The standard protocol starts with a TAS with the full bladder, serving as an acoustic window. Following bladder emptying, the Patient assumes the lithotomy position and TVS is performed. The two imaging techniques are complementary and often provide different diagnostic information. Color, power and spectral Doppler are usually performed to assess the grayscale findings better. The 3D ultrasound is helpful in assessing the position of the IUD to better evaluate submucosal leiomyomas and the bottom of the uterus as well as foetal morphology.	Wider field of view than TVS; Better evaluation of the superficial and distal structures of the vagina by bringing the probe closer to the target organ.	Empty bladder; Obese patients; Retroverse uterus where the fundus is located beyond the focal zone of the transducer; Less accurate for characterization of adnexal masses.
**Transvaginal sonography (TVS)**	High-frequency probe (endocavitary probe > 7 MHz).		High spatial resolution in assessing uterus, ovary, and adnexal structures.	Limited field of view. TVS should not be performed on patients who are unable or unwilling to give consent to the procedure, virgin patients and if the insertion of the probe produces considerable discomfort. Contraindicated during the second and third trimesters of pregnancy if there is the risk of active bleeding or membrane rupture.

## Data Availability

Data sharing is not applicable.

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
