# Peer review of "Role of Ultrasound in the Assessment and Differential Diagnosis of Pelvic Pain in Pregnancy"

_diagnostics, 2022, doi:10.3390/diagnostics12030640_

Round 1

Reviewer 1 Report

Dear Authors,

thanks for submitting this interesting pictorial review on the use of ultrasound in differentiating causes of pelvic pain in pregnancy.

The paper reads well and is of interest to the general audience, particularly to the radiologist or gynaecologist approaching ultrasound for the first time.

Aside from minor grammatical errors, I have no opposition to its publication. 

Author Response

Thank you very much for your positive comments.

We have made an extensive grammatical revision of the manuscript.

Reviewer 2 Report

A very good and analytical review. The authors have extensively described the use of ultrasound in diagnosis of pelvic pain during pregnancy. I have one question, the use of abdominal and transvaginal scan is well described and know. Do you have any reports regarding the use of transperineal scan in acute pelvic conditions during pregnancy ? 

Author Response

Thank you very much for your review.

Unfortunately, we have no experience about the use of transperineal scan in acute pelvic conditions during pregnancy.
